# Glucose Hydrogenolysis into 1,2-Propanediol Using a Pt/deAl@Mg(OH)_2_ Catalyst: Expanding the Application of a Core–Shell Structured Catalyst

**DOI:** 10.3390/nano12213771

**Published:** 2022-10-26

**Authors:** Shizhuo Wang, Jikang Jiang, Minyan Gu, Yuanbo Song, Jiang Zhao, Zheng Shen, Xuefei Zhou, Yalei Zhang

**Affiliations:** State Key Laboratory of Pollution Control and Resources Reuse, Key Laboratory of Yangtze River Water Environment of MOE, College of Environmental Science and Engineering, Tongji University, Shanghai 200092, China

**Keywords:** 1,2-propanediol, alkaline modification, core–shell catalyst

## Abstract

To substitute fossil resources, it is necessary to investigate the conversion of biomass into 1,2-propanediol (1,2-PDO) as a high-value-added chemical. The Pt/deAl-Beta@Mg(OH)_2_ catalytic system is designed to obtain a higher 1,2-PDO production yield. The optimal yield of 1,2-PDO is 34.1%. The unique shell-core structure of the catalyst demonstrates stability, with a catalytic yield of over 30% after three times of use. The primary process path from glucose to 1,2-PDO, glucose-hexitol-1,2-PDO, is speculated by the experiments of intermediate product selectivity. The alkaline catalytic mechanism of the reaction process is elucidated by studying catalyst characterization and analyzing different time courses of products. The introduction of Mg(OH)_2_ improves the target yield by promoting the isomerization from glucose to fructose and retro-aldol condensation (RAC) conversion, with pseudo-yield increases of 76.1% and 42.1%, respectively. By studying the processes of producing lactic acid and 1,2-PDO from glucose, the glucose hydrogenolysis flow chart is improved, which is of great significance for accurately controlling 1,2-PDO production in industrial applications. The metal, acid, and alkali synergistic catalytic system constructed in this paper can provide a theoretical basis and route reference for applying biomass conversion technology in practice.

## 1. Introduction

1,2-PDO is an environmentally friendly intermediate chemical synthesis with many applications [1,2]. As glucose is an extremely important biomass model compound, studying its conversion into 1,2-PDO will help develop and utilize essential biomass resources. The hydrogenolysis of glucose is complicated and accompanied by many by-products, e.g., furan by-products from dehydration and other by-products from hydrogenation [3,4]. In order to improve the production yield of 1,2-PDO, the construction of a catalyst system that can enhance the conversion efficiency of glucose isomerization and RAC has been the focus of recent studies [5,6].

As a solid catalyst for heterogeneous catalytic systems, beta zeolite has been widely used in biomass catalysis, adsorption, and other fields [7]. Sn-Beta zeolites prepared by hydrothermal synthesis are used to catalyze the conversion of sugars to lactic acid or lactic acid esters in aqueous and organic phases [8]. Modified beta zeolite immediately triggered its research boom in biomass catalytic utilization due to its excellent catalytic performance and structural stability as a heterogeneous catalyst. In the preparation of 1,2-PDO, the 6C bond cleavage of glucose and fructose is based on the retro-aldol reaction (RAC). Zhang designed a Pb-Sn/β zeolites catalytic system and studied the effects on glucose conversion. The results suggest that the introduction of Sn promotes the hydrolysis of glucose by enhancing the RAC of glucose and the isomerization from glucose to fructose [9]. Among the modifications of beta zeolites, dealumination and metal modifications are the most widely used methods. The beta zeolites modified by dealumination can obtain the appropriate Brønsted acid site and Lewis acid site and increase the service life of the catalysts. Furthermore, the metal modification is more dependent on the dispersion ability and acidity of the loading metals. Pt is a commonly used metal for beta zeolite modification, which affects the selectivity and yield of the product by acidity site distribution and pore volume (applications shown in Appendix A). Lactic acid (3C sugar), one of the essential platform compounds, can be converted into many valuable chemicals, such as acetaldehyde, 1,2-PDO, and propionic acid. In our previous study, the catalytic system of Zn-Sn/β zeolites was prepared to realize a one-pot selective conversion of glucose for producing lactic acid [10]. The introduction of Zn makes the catalyst possess alkaline properties. During the reaction, the catalyst neutralizes the acid, reducing the effect of the acid on fructose dehydration, which confirms that the alkali modification of the beta zeolite is necessary for converting glucose. The 1,2-PDO can be quickly produced from lactic acid after hydrogenation. Therefore, for the large-scale industrial production of 1,2-PDO and further exploration to improve the yield, it is necessary to study the differences between the reaction mechanisms of producing lactic acid and 1,2-PDO catalyzed by the beta zeolites.

In this paper, we designed a Pt/deAl-Beta@Mg(OH)_2_ alkaline core–shell structure catalytic system for producing 1,2-PDO. The factors, including dealumination time, metal loading, alkali loading, reaction time, reaction temperature, and catalyst dosage, were optimized, and the mechanism of alkali modification was explored by the results of catalyst characterization. Based on the acid–base analysis, the effect of different catalyst modifications for the reaction routes is discussed for controlling the selectivity of 1,2-PDO preparation. According to the above explorations, the catalytic reaction system for improving the production of 1,2-PDO will be constructed, and the possible reaction routes and mechanism of catalytic reaction will be proposed.

## 2. Materials and Methods

### 2.1. Chemicals

The glucose (99.5%), fructose (99%), ethylene glycol (EG, 99%), MgO (98.5%), and chloro-platinic acid (H_2_PtCl_6_·6H_2_O, 1000 mg/L) were purchased by Sigma Aldrich (St. Louis, MO, USA). The 1,2-PDO (99.5%), 1,3-dihydroxyacetone (1,3-DHA, 99%), 1,2-butanediol (1,2-BDO, 99%), hexitol (sorbitol and mannitol, 98%), 1,2-hexanediol (1,2-HDO, 98%), and acetol (99.5%) were supplied by Aladdin Bio-Chem Technology Co., Ltd. (Shanghai, China). All chemicals were at least analytical grade.

### 2.2. Catalyst Preparation

The alkaline core–shell structure with different Mg(OH)_2_ nominal weight loadings in the range of 0–100% (x wt%) was prepared by three steps, including (1) preparing the dealuminated Beta (deAl-Beta), (2) modifying the deAl-Beta with Pt and (3) hydrothermal treatment for the concurrent accomplishment of Mg(OH)_2_ assembly onto Pt/deAl-Beta.

#### 2.2.1. deAl-Beta Preparation

The beta zeolite and concentrated nitric acid were added to a three-necked round-bottom flask in 1 g:20 mL. Then the flask was placed in an oil bath at a controlled temperature of 80 ℃, and the alumina was removed by condensation reflux at a stirring rate of 200 rpm for a specific time (0, 10, 20 h). The solid–liquid mixture was separated by centrifugation at 3000 rpm for 20 min in a high-speed centrifuge. The nitric acid residue on the beta zeolite was then washed with deionized water until the supernatant was pH neutral. The cleaned solid fraction was dried in an oven at 150 °C. The solid powder was called dealuminated beta zeolite, denoted as deAl-Beta [11].

#### 2.2.2. Pt/deAl-Beta Preparation

Pt (1~5 wt%) was introduced into the structure of deAl-Beta by isovolumetric impregnation using H_2_PtCl_6_·6H_2_O as the precursor. H_2_PtCl_6_·6H_2_O was dissolved in a certain (10 mL) amount of deionized water to prepare a uniform solution. The solid carrier was added into the above H_2_PtCl_6_·6H_2_O solution and stirred evenly before ultrasonic treatment for 15 min. Then, the sample was placed in room temperature air for 6 h and dried at 105 °C for 6 h. Finally, the sample was roasted in a tube furnace for 4 h at 450 °C with stagnant air and a temperature rise rate of 2 °C/min. The metal-modified deAl-Beta was prepared as Pt/deAl-Beta.

#### 2.2.3. Pt/deAl-Beta@Mg(OH)_2_ Preparation

A certain amount of catalyst (50~300 mg, details shown in *3.4*), 10 mL deionized, and MgO (0~75 wt%, gradient sampling shown in *3.3*) precursor was added to the reaction kettle. The MgO was hydrothermally treated at 5 M Pa hydrogen pressure and 200 °C for 4 h. Then, the solid–liquid mixture was separated by a high-speed centrifuge (4000 rpm, 4 min). The solid obtained by centrifugation was dried overnight in an oven at 150 °C and then placed in an agate mortar. After uniform mixing and grinding for 30 min, the fully ground solid powder was transferred to a crucible and placed in a tube furnace at 550 °C. The solid powder was calcined in flowing air for 6 h to remove impurities, and finally, the Pt/deAl-Beta@Mg(OH)_2_ catalyst was obtained.

### 2.3. Catalyst Characterization

The catalyst was analyzed by the following methods: nitrogen adsorption–desorption curve determination, X-ray diffraction (XRD, Bruker D8), temperature programmed desorption of carbon dioxide (CO_2_-TPD, Micromeritics AutochemII 2920), temperature programmed desorption of ammonia (NH_3_-TPD), scanning electron microscopy/transmission electron microscopy (SEM/TEM) and pyridine adsorption infrared spectroscopy (Py-IR). The catalyst was analyzed by gas chromatography (GC, Agilent 7820A) and liquid chromatography (LC, Agilent 1200).

The specific surface area and pore size distribution of catalysts were analyzed by Nitrogen adsorption–desorption (Micromeritics ASAP 2020M) [12]. Each sample was purged in a vacuum at 300 °C for 3 h prior to measurement. The specific surface area of the catalysts was calculated using the Brunauer–Emmett–Teller (BET, Perkin Elmer Optima 2100 DV) method, and the pore volume was calculated using the Barrett–Joyner–Halenda (BJH) model [13]. The crystal structure of the catalysts was characterized by X-ray diffraction using a Bruker D8 radiometer (40 kV, 40 mA, CuK radiation of = 1.54 Å) at room temperature with a scan speed of 2°/min, a scan step of 0.02°, and a 2θ angle range of 5~90°. The acidity and basicity of the catalysts were determined by NH_3_-TPD and CO_2_-TPD, respectively. The catalyst was dried in an argon atmosphere to remove surface gas components and water, and then saturated NH_3_ and CO_2_ were adsorbed at room temperature, followed by a programmed temperature rise at 10 °C/min at an argon flow rate of 30 cm^3^/min and a TCD detector to record the carrier gas desorption from 45 °C to 800 °C. The catalyst was observed by SEM (Hitachi S4800), and TEM (JEOLJEM-1230) images of the catalyst were recorded to observe the morphological characteristics of the catalyst. The acidic species and the number of acidic sites of the catalysts were determined using a Fourier infrared spectrometer (Perkin Elmer Frontier, FT-IR). The samples were dried under vacuum at 450 °C for 2 h before testing and then cooled to room temperature. The pyridine vapor was introduced into the sample chamber and fully adsorbed pyridine for 0.5 h. The sample chamber was then controlled to warm up to 150 °C, 250 °C, 350 °C, and 450 °C for 1 h at each temperature. The sample chamber was stabilized for 1 h, and the amount of pyridine was quantified in IR [14].

After each reaction, the solid–liquid mixture in the reactor was taken out and centrifuged. The solid was washed with anhydrous ethanol and then calcined in a 550 ℃ tube furnace for 6 h for regeneration. The prepared catalyst was recycled in this way.

### 2.4. Catalytic Reaction

The catalytic experiments were carried out in a 50 mL stainless steel autoclave. The catalyst reduction and Mg(OH)_2_ in situ hydrothermal encapsulation reactions were first performed. Then, 200 mg of catalyst, a certain amount of MgO precursor, 10 mL of water, and a magnetic stirrer were added to the reactor. The reactor was vented three times using 5 M Pa of hydrogen [15]. After hydrothermal treatment at 200 °C and 600 r/min for 4 h, the reactor was naturally cooled to room temperature. A certain amount of glucose was added to the reaction system, and the reactor was resealed. The reaction was carried out at a certain temperature. The liquid phase product and the solid phase catalyst were collected separately. The liquid phase products were quantified using gas chromatography (Agilent 7820A) with an FID detector on a DB-WAXetr (30.0 m × 530 µm × 1 µm) column. The solid catalyst was characterized, cleaned, and stored for next use. The GC and HPLC operating conditions are as follows.

#### 2.4.1. GC Operating Condition

Chromatographic column: DB-WAXetr

Sample volume: 0.4 μL

Injector temperature: 240 °C

Detector temperature: 250 °C

Column temperature: from 60 °C to 85 °C (5 ℃/min), to 210 °C (15 °C/min)

and then to 230 °C (8 °C/min), hold for 3 min.

Carrier gas flow rate: 20 mL/min

#### 2.4.2. HPLC Operating Condition

Chromatographic column: Shodex SUGAR SH1011

Sample volume: 10 μL

Mobile phase: 0.005 M H_2_SO_4_

Flow rate: 0.5 mL/min

Column temperature: 55 °C

### 2.5. Calculation

The glucose conversion (%) was defined as
(1)$$\mathrm{Conversion}(\%)=\frac{\mathrm{moles}\;\mathrm{of}\;\mathrm{glucose}\;\mathrm{after}\;\mathrm{reaction}}{\mathrm{moles}\;\mathrm{of}\;\mathrm{glucose}\;\mathrm{before}\;\mathrm{reaction}}\times 100\%$$

The yield of products (%) was defined as
(2)$$\mathrm{Yield}(\%)=\frac{\mathrm{the}\;\mathrm{number}\;\mathrm{of}\;\mathrm{carbon}\;\mathrm{atoms}\;\mathrm{in}\;\mathrm{the}\;\mathrm{product}\;\mathrm{molecule}\times \mathrm{moles}\;\mathrm{of}\;\mathrm{the}\;\mathrm{product}}{6\times \mathrm{moles}\;\mathrm{of}\;\mathrm{glucose}\;\mathrm{before}\;\mathrm{reaction}}\times 100\%$$

The selectivity of products (%) was defined as
(3)$$\mathrm{Selectivity}(\%)=\frac{\mathrm{The}\;\mathrm{yield}\;\mathrm{of}\;\mathrm{the}\;\mathrm{product}}{\mathrm{Conversion}}\times 100\%$$

The pseudo-yield of isomerization and RAC reaction (%) was defined as [16]
(4)$$\mathrm{Pseudo}-\mathrm{yield}\;\mathrm{of}\;\mathrm{isomerization}(\%)=\frac{\mathrm{Yield}\;\mathrm{of}\;1,2-\mathrm{PDO}\;\mathrm{from}\;\mathrm{glucose}}{\mathrm{Yield}\;\mathrm{of}\;1,2-\mathrm{PDO}\;\mathrm{from}\;\mathrm{fructose}}\times 100\%$$
(5)$$\mathrm{Pseudo}-\mathrm{yield}\;\mathrm{of}\;\mathrm{RAC}(\%)=\frac{\mathrm{Yield}\;\mathrm{of}\;1,2-\mathrm{PDO}\;\mathrm{from}\;\mathrm{fructose}}{\mathrm{Yield}\;\mathrm{of}\;1,2-\mathrm{PDO}\;\mathrm{from}\;\mathrm{DHA}}\times 100\%$$

## 3. Results

### 3.1. Effect of Dealumination Time

In order to investigate the effect of the dealumination time of beta zeolite on the hydrogenolysis of glucose, the catalytic effects of dealumination for 10 h and 20 h and the control group (without dealumination) on glucose are explored. The conversion of EG and 1,2-PDO with Pt/deAl-Beta_(20)_@Mg(OH)_2_ is better than the other two catalysts (Figure 1a). The highest 1,2-PDO yield of 34.1% is obtained after 3 h catalytic reaction, which proves that dealumination can effectively improve the selectivity of glucose hydrogenolysis to 1,2-PDO (Figure 1b). Thus, the Pt/deAl-Beta_(20)_@Mg(OH)_2_ catalytic system is selected for optimization. If there is no particular explanation, it is referred to as Pt/deAl-Beta@Mg(OH)_2_ below.

### 3.2. Effect of Pt Loading

In the previous study of Pt modification on beta zeolite for promoting catalytic efficiency, the Pt loading was less than 1.0 wt% [17,18,19,20,21]. However, without Mg(OH)_2_ loading, the induction of Pt inhibits the production of 1,2-PDO in the designed catalysis system. The catalytic performance of the catalysts with different Pt loadings of Mg(OH)_2_ (7.5 wt%) and without Mg(OH)_2_ was explored using glucose as the substrate (Figure 2) [22]. It can be inferred that the loading of Mg(OH)_2_ dramatically improves the yield of 1,2-PDO. The yield accelerates from 4.5% to 34.1% with Pt loading of 3.0 wt%, increasing approximately eight times. As for Pt/deAl-Beta, there is almost no effect of the loading amount of Pt on the yield of 1,2-PDO, maintaining at approximately 5.0%. With loading the Mg(OH)_2_, the optimal loading of Pt was reduced by 2%, and the production yield increased by 29.31%, confirming that the Pt usage was reduced. The yield of 1,2-PDO gradually rises with the improvement of Pt load and reaches the maximum with the Pt load of 3.0 wt%. When the load exceeds 3.0 wt%, the catalytic ability gradually decreases. Thus, the optimal Pt load concentration is 3.0 wt%.

### 3.3. Effect of Mg(OH)_2_ Loading

The effect of alkali modification by loading Mg(OH)_2_ on the catalyst system was studied after dealumination modification and metal modification. When the loading amount of Mg(OH)_2_ is 2.5 wt%, the yield of 1,2-PDO promotes rapidly (Figure 3a). The increase is approximately six times that without Mg(OH)_2_ loading, suggesting that the loading of Mg(OH)_2_ indeed improves the formation of 1,2-PDO. Then, the yield of 1,2-PDO advances gradually with the enhancement of the Mg(OH)_2_ loading until the yield of Mg(OH)_2_ loading of 7.5 wt% reaches the maximum value, and the yield of 1,2-PDO is approximately 34.1% (Figure 3a). The loading of Mg(OH)_2_ begins to decrease the production of 1,2-PDO at around 12.5 wt%, where the yield of 1,2-PDO is at a high level of 32.7%. With the improvement in alkali loading, the selectivity of 1,2-HDO decreases significantly, which confirms that the loading of Mg(OH)_2_ could effectively inhibit the formation of 1,2-HDO and promote the selectivity of 1,2-PDO conversion (Figure 3b). However, the yield reduces after a large amount of loading. The excessive loading of Mg(OH)_2_ may not be conducive to converting DHA to 1,2-PDO. In summary, the optimal loading of the Mg(OH)_2_ catalyst is 7.5 wt%.

### 3.4. Effect of Catalyst Dosage

When the dosage of the catalyst rises from 50 mg to 200 mg, the yield of 1,2-PDO advances significantly from 18.3% to 34.1% (Figure 4). After that, the yield of 1,2-PDO decreases with the continued improvement in catalyst dosage. On the one hand, if the catalyst dosage is insufficient, it cannot provide enough acidic sites. On the other hand, if the catalyst dosage is too much, the excessive Mg(OH)_2_ will affect the balance of the reaction, e.g., not conducive to the conversion of DHA to 1,2-PDO or not conducive to the formation of 1,2-PDO [22]. From the perspective of economic feasibility, the optimal catalyst dosage is 200 mg under reaction conditions.

### 3.5. Effect of Catalytic Reaction Conditions

When the temperature rises from 180 °C to 200 °C, the yield of 1,2-PDO improves to a certain extent, and the maximum yield reaches 34.1% at 200 °C after 3 h of reaction. When the temperature increases from 220 °C, the peak yield of 1,2-PDO decreases significantly (Figure 5a). With the improvement of the reaction process and the temperature, the yield of EG advances tardily, remaining between 5.0% and 8.0% (Figure 5b). The promotion effect of improving the temperature on the formation of NPA is the strongest (Figure 5c). It is because, at high-temperature conditions, 1,2-PDO continues to decompose into large amounts of EG and NPA, which reduces the yield of 1,2-PDO. Therefore, the optimal reaction conditions in the catalytic system are a temperature of 200 °C and a time of 3 h.

### 3.6. Catalyst Reusability

In the second and third catalytic reactions, the yield of 1,2-PDO reduces slightly from 34.1% to 33.5% and 32.5% (Figure 6). The unique core–shell structure can effectively prevent the loss of supported metals and promote the structural stability of the catalyst. The Pt content of the catalyst decreases slightly from 3.0 wt% to 2.8 wt% and 2.7 wt% in three reuses, respectively. Thus, the encapsulation of the Mg(OH)_2_ shell effectively prevents the loss of Pt and makes the structure more stable in the catalytic reaction. The catalyst shows good recycling performance. Moreover, a similar effect of a solid base on catalytic stability was also reported by Sun [23].

## 4. Discussion

### 4.1. Characterization of Catalysts

#### 4.1.1. TEM Analysis

The Pt/deAl-Beta catalyst prepared by the equal volume impregnation method has an excellent loading effect (Figure 7b). Pt particles are uniformly distributed on the deAl-Beta support, with sizes of about 5 nm. Compared with Pt/deAl-Beta, the Pt particles in the Pt/deAl-Beta@Mg(OH)_2_ catalyst prepared by in situ hydrothermal synthesis has a higher dispersion and smaller particle size (Figure 7c). After the hydrothermal reaction, the magnesium species is mainly wrapped on the surface of Pt/deAl-Beta in the form of a shell with a thickness between 10 and 40 nm. Due to the existence of a large number of free hydroxyl groups under hydrothermal conditions, the MgO precursor could be transformed into Mg(OH)_2_ in large quantities, which attaches to the catalyst surface in the form of a shell (Figure 7d).

#### 4.1.2. BET Analysis

There are two hysteresis loops at the relative pressure of P/P_0_ < 0.01 and 0.4 < P/P_0_ < 0.9 of the Pt/deAl-Beta and Pt/deAl-Beta@Mg(OH)_2_ (Appendix A). Corresponding to type I and type IV isotherms, mesoporous pores and mesoporous channels exist in the catalysts. After alkali modification, the hysteresis loop of Pt/deAl-Beta becomes more significant and positively correlated with the Mg(OH)_2_ loading. Moreover, the number of mesoporous channels or pore volume increases significantly [24]. This confirmed that the Mg(OH)_2_ contributes to the formation of mesoporous channels and the dispersion of Pt particles (Appendix A). In the research of the Ni/ZSM-5 molecular sieve catalyst, the reduction in pore volume and specific surface area can improve the selectivity of the molecular sieve and decrease the formation of by-products. The promotion of strong acidic sites also advances the selectivity of target products [25]. The alkali modification increases the selectivity of catalytic reaction in the structure base.

#### 4.1.3. XRD Analysis

The characteristic peak at about 2θ = 22.5° suggests that the beta zeolite treated by concentrated nitric acid dealumination modification, high temperature roasting metal loading, and alkali modification do not undergo severe skeleton collapse [26]. The characteristic peaks of 2θ = 39.8°, 46.2°, and 67.5° correspond to the (111), (200), and (220) crystal planes in the JCPDS No. 04-0802 library, indicating that the platinum nanoparticles are successfully loaded onto the catalyst. After loading Mg(OH)_2_, the half-peak width of Pt (111) and (200) in the catalyst increases, which confirms that the Mg(OH)_2_ shell can effectively prevent the agglomeration of Pt nanoparticles under hydrothermal conditions and promote the dispersion of Pt nanoparticles [27]. There is no peak of Mg(OH)_2_ in the XRD analysis, which suggests that Mg(OH)_2_ covers the surface of the molecular sieve in an amorphous manner (Appendix A). The Mg(OH)_2_ shell structure does not entirely block the reaction sites of Pt nanoparticles, which explains why the Pt/deAl-Beta@Mg(OH)_2_ catalytic system has both acid and alkali catalytic ability.

#### 4.1.4. Acid–Base Analysis

The Pt/deAl-Beta@Mg(OH)_2_ catalyst has a unique shell–core structure, which combines the characteristics of Pt/deAl-Beta and Mg(OH)_2_. So, it has rich acidic and alkaline sites. There are almost no alkaline sites on the beta zeolite after dealuminization. Loading Pt makes no effect on the alkalinity of the catalyst (Appendix A). Generally, the weakly acidic and alkaline sites occur at adsorption temperatures below 200 °C, the medium acidity and medium alkaline sites occur at temperatures between 200 and 400 °C, and the strong acidity and alkaline sites occur at temperatures over 400 °C [28]. The weak and strong alkaline sites of Pt/deAl-Beta are significantly increased after alkali modification, confirming that the Mg(OH)_2_ has been successfully loaded onto the surface of the catalyst (Appendix A).

The loading of Mg(OH)_2_ changes the distribution of acidity sites in Pt/deAl-Beta, resulting in a shift from weak acidity sites to medium and strong acidity sites (Appendix A). The acidity sites of Pt/deAl-Beta and Pt/deAl-Beta@Mg(OH)_2_ are mainly Lewis acids (Table 1). Only if the desorption temperatures are 150 °C and 250 °C will a small amount of Brønsted acidity sites appear. After loading Mg(OH)_2_, the number of strong acidity sites increases, but the amount of weak acidity and the total amount of acidity decreases. Overall, Mg(OH)_2_ does not affect the total amount of acid sites too much, but the distribution of acidity sites changes from weak to strong (Appendix A).

Combined with the results of Pt loading, the Pt/deAl-Beta catalyst inhibits the isomerization of glucose to fructose due to the lack of the essential sites required for catalyzing the reaction, resulting in a low overall catalytic capacity. At the same time, the Pt/deAl-Beta@Mg(OH)_2_ catalytic system has sufficient acid and alkaline sites. The hydrogen spillover effect of Pt effectively improves the hydrogenation ability of the catalyst, strengthening the catalytic ability [29]. However, too much Pt loading can easily enhance the agglomeration, and too much Lewis acidity will lead to RAC of non-isomerized glucose to produce other products, which is not conducive to the formation of 1,2-PDO (shown in the second paragraph of *4.3*).

### 4.2. Analysis of Alkaline Catalytic Reaction Mechanism

The catalytic selectivity of possible intermediate products and reaction routes were analyzed (Figure 8). In order to further determine the mechanism whereby alkali modification promotes the yield of 1,2-PDO under the optimal conditions of the system, glucose, fructose, and DHA were used as substrates for catalytic reactions (Appendix A). The glucose conversion process of the catalytic system without Mg(OH)_2_ is relatively slow, and the reaction completes in approximately 30 min. After loading Mg(OH)_2_, the early conversion process of glucose is very rapid, and the reaction completes at approximately 0 min. When the Mg(OH)_2_ is not loaded, hexitol is generated in large quantities before 120 min of reaction and then decreases sharply, while 1,2-HDO is generated. This suggests that 1,2-HDO is obtained by the hydrogenation of glucose and fructose. With the Mg(OH)_2_ loaded, the concentration of 1,2-HDO remains at a low level, confirming that Mg(OH)_2_ can inhibit the side reaction of the conversion of hexitol to 1,2-HDO (Figure 9. Route 2). Furthermore, the selectivity of DHA to 1,2-PDO is decreased by over 40%, but with the glucose as a substrate, the yield of 1,2-PDO is increased by 29.8%, proving that Route 3 is the primary path for producing 1,2-PDO (Figure 9. Route 3). The concentration of fructose increases first and then decreases, and the final concentration decreases to 0, confirming that fructose is steeply converted to DHA by RAC after glucose isomerization. In the catalytic system loaded with Mg(OH)_2_, the peak yield of fructose is 5.2-times higher than that without Mg(OH)_2_, suggesting that the loading of the Mg(OH)_2_ system promotes the efficiency of the glucose isomerization process. The rate of fructose formation and conversion per unit time is 4.7-times and 5.2-times higher than without loading Mg(OH)_2_. The peak yield of DHA is 3.73-times higher than that without loading, indicating that Mg(OH)_2_ can improve the RAC of fructose. When fructose is used as the substrate, the conversion rate of fructose to hexitol is increased after loading Mg(OH)_2_. The isomerization pseudo-yield of glucose to fructose increases from 12.2% to 88.3%, which suggests that the Mg(OH)_2_ significantly promotes the isomerization process. With fructose as the substrate, the selectivity conversion of 1,2-PDO advances from 35.4 wt% to 38.6 wt% after loading Mg(OH)_2_, which proves that the Mg(OH)_2_ not only improves the isomerization of glucose to fructose but also further increases the fructose RAC. Further comparing the yield of 1,2-PDO with DHA and fructose as substrates, the pseudo-yield of RAC rises from 40.3% to 82.4% after loading Mg(OH)_2_, confirming that Mg(OH)_2_ accelerates the RAC and the directional bond breaking of fructose [8]. With sorbitol and mannitol as substrates, the selectivity of 1,2-PDO conversion yield increases from 0.0% to 68.7% and 58.6%, respectively. In summary, the Mg(OH)_2_ increases the yield of 1,2-PDO by promoting isomerization, RAC, and the conversion of hexitol. The temperature changes in the heating process before time 0 were shown in Appendix A.

### 4.3. Comparison of Two Catalytic Mechanisms

The conversion of lactic acid to 1,2-PDO is facilitated after hydrogenation. For completing the catalytic reaction system for producing 1,2-PDO, the differences between the preparation routes and reaction mechanisms of lactic acid and 1,2-PDO were further studied. According to the lactic acid production system, the proposed reaction routes for the formation of different products from glucose were added to Figure 9 [30].

As for lactic acid, glucose is isomerized to fructose under the action of Lewis acid (both Zn and Sn can provide Lewis acid sites). Then fructose was converted to DHA or glyceraldehyde by RAC through promoting the C_3_-C_3_ directional bond breaking of fructose [31]. These two substances continued to produce methylglyoxal by dehydration and then isomerization to lactic acid (Figure 9, Route 5). Sn plays a crucial role in these two steps by providing Lewis acid sites. Fructose is prone to dehydration to produce 5-hydroxymethylfurfural (5-HMF) under acidic substances. Nevertheless, introducing Zn into beta zeolites increases the alkalinity, weakening the dehydration of fructose. In addition, 5-HMF is an active chemical that can continue to be hydrated in acidic solutions to produce formic acid, levulinic acid, or other products. The presence of Zn can effectively reduce the decomposition and conversion of 5-HMF by providing an alkaline environment.

As for 1,2-PDO, the acid site distribution of beta zeolite is optimized by three methods: dealumination, metal, and alkali modification. Through in situ hydrothermal synthesis, Mg(OH)_2_ forms a porous shell structure on the surface of the beta zeolite and provides an alkaline environment. In particular, the effect of alkali modification on the production yield is much more significant than that of the other two modification methods. The yield of 1,2-PDO on the catalyst loaded with Mg(OH)_2_ is approximately eight times that of the catalyst without loading. According to the product analysis, almost no lactic acid is detected. It is because the lactic acid is easily hydrogenated to produce 1,2-PDO in a hydrogen atmosphere. The catalytic mechanism of alkali modification was discussed. On the one hand, the loading of Mg(OH)_2_ inhibits the dehydration reaction of fructose to produce 5-HMF (Figure 9. Route 6) by neutralizing the acidity. The selectivity conversion of DHA to 1,2-PDO through hydrogenation is reduced by 40.5% (Figure 9, Route 4). On the other hand, Mg(OH)_2_ promotes the hydrogenation reaction and improves the yield of 1,2-PDO through the conversion of hexitol (Figure 9, Route 3), with the selectivity conversion increase of 58.6% of sorbitol and 68.7% of mannitol. In addition, the loading of Mg(OH)_2_ improves the isomerization of glucose to fructose and the fructose RAC, increasing the pseudo-yield by 76.1% and 42.1%, respectively. Route 3, considered the main reaction path, dramatically increases the target production yield. It provides a new idea for the efficient preparation of 1,2-PDO. The determination of the reaction path also provides a reference for the subsequent expansion of the system, preparing other high-value-added compounds with glucose.

## 5. Conclusions

In this paper, a Pt/deAl-Beta@Mg(OH)_2_ catalytic system with a core–shell structure was constructed to obtain a higher yield of 1,2-PDO, and its catalytic performance for the conversion of glucose to 1,2-PDO was explored. The optimal conditions for 1,2-PDO production are determined as follows: Pt loading 3.0 wt%, Mg(OH)_2_ loading 7.5 wt%, catalyst dosage 200 mg, glucose solution 10 mL (11.25 mg/mL), reaction temperature 200 °C, and reaction time 3 h. The yield of 1,2-PDO is 34.1%. After three-time reactions, the catalyst demonstrates stability, with a production yield of at least 30%. The dealumination and metal modification treatments of the catalysts provide many Lewis acid sites. However, among the modification methods prepared in the catalytic system, the effect of the loading of Mg(OH)_2_ is the most significant, with the catalytic efficiency of loaded Mg(OH)_2_ being approximately eight times that of unloaded Mg(OH)_2_. Therefore, the reinforcement of the catalytic reaction mechanism of alkali modification was analyzed with products of different time courses of glucose hydrogenolysis. Mg(OH)_2_ contributes to facilitating the production of 1,2-PDO through isomerization and RAC of glucose, increasing the pseudo-yield by 76.1% and 42.1%, respectively. Furthermore, the major process path of glucose to 1,2-PDO, glucose-hexitol-1,2-PDO is speculated by the experiments of intermediate product selectivity. In the beta zeolite catalytic system, the preparation routes of lactic acid are similar to 1,2-PDO, and the latter can be easily obtained by lactic acid hydrogenation. By studying the processes of producing lactic acid and 1,2-PDO from glucose, the flow chart of glucose conversion in beta zeolite is improved. It will facilitate an alternative to the traditional technological route of 1,2-PDO based on fossil fuels and help mitigate the current looming climate warming and energy crisis.

## Figures and Tables

**Figure 1 nanomaterials-12-03771-f001:**
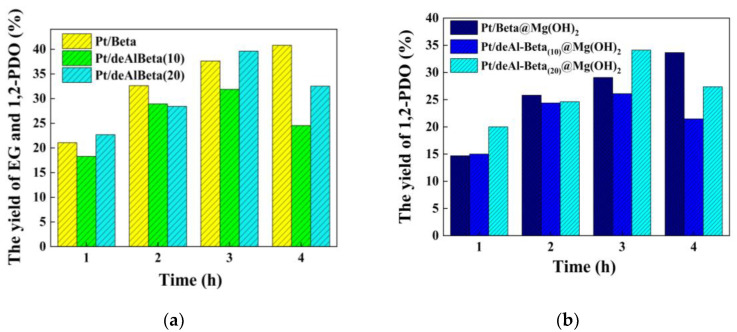
The effect of beta zeolite dealumination time on glucose hydrogenolysis into (**a**) 1,2-PDO and EG; (**b**) 1,2-PDO. Reaction condition: catalyst (200 mg), Pt loading (3.0 wt%), Mg(OH)_2_ loading (7.5 wt%), glucose (11.25 mg/mL, 10 mL) and H_2_ (initial 6 MPa) at 200 ℃ for 3 h.

**Figure 2 nanomaterials-12-03771-f002:**
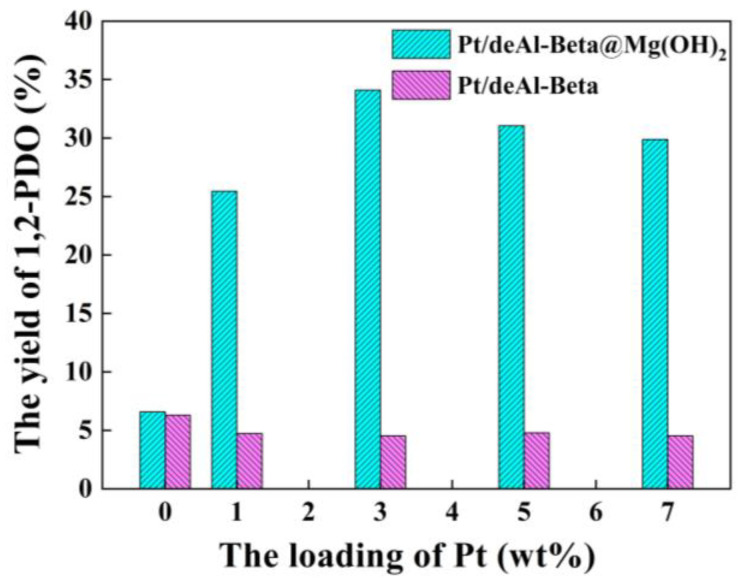
The effect of Mg(OH)_2_ and Pt loading on glucose hydrogenolysis into 1,2-PDO. Reaction condition: catalyst (200 mg), glucose (11.25 mg/mL, 10 mL) and H_2_ (initial 6 MPa) at 200 °C for 3 h.

**Figure 3 nanomaterials-12-03771-f003:**
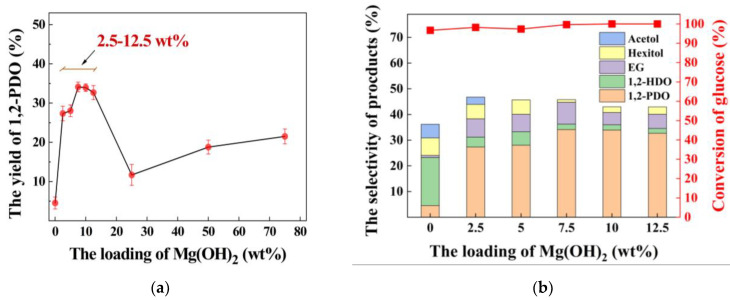
The effect of Mg(OH)_2_ loading on hydrogenolysis of glucose into 1,2-PDO. (**a**) effect of Mg(OH)_2_ loading(0–80 wt%) on 1,2-PDO production; (**b**) effect of Mg(OH)_2_ loading on 1,2-PDO production (0–12.5 wt%) and selectivity of production. Reaction condition: catalyst (200 mg), glucose (11.25 mg/mL, 10 mL) and H2 (initial 6 MPa) at 200 °C for 3 h.

**Figure 4 nanomaterials-12-03771-f004:**
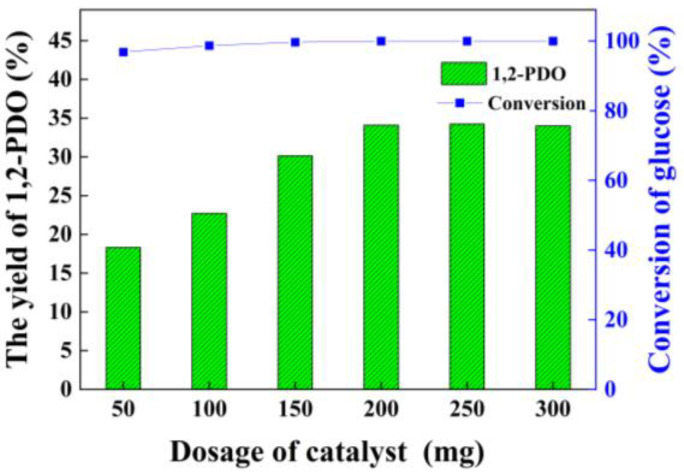
The effect of catalyst dosage on glucose hydrogenolysis into 1,2-PDO. Reaction condition: Pt loading (3.0 wt%), glucose (11.25 mg/mL, 10 mL) and H_2_ (initial 6 MPa) at 200 °C for 3 h.

**Figure 5 nanomaterials-12-03771-f005:**
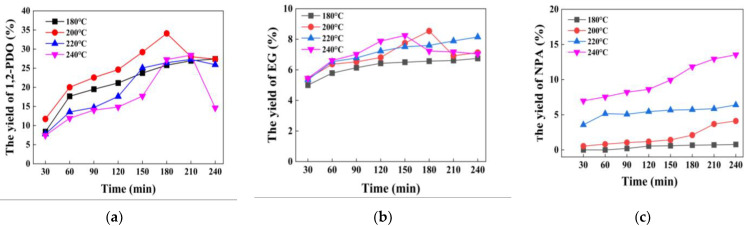
The effect of reaction temperature and reaction time on glucose hydrogenolysis into (**a**) 1,2-PDO; (**b**) EG; (**c**) NPA. Reaction condition: catalyst (200 mg), glucose (11.25 mg/mL, 10 mL) and H_2_ (initial 6 M Pa).

**Figure 6 nanomaterials-12-03771-f006:**
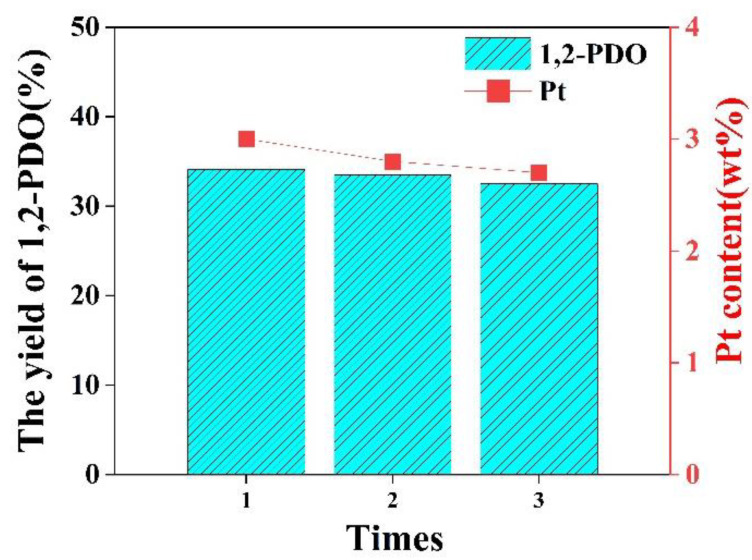
Reusability and metal leaching of Pt/deAl-Beta@Mg(OH)_2_.

**Figure 7 nanomaterials-12-03771-f007:**
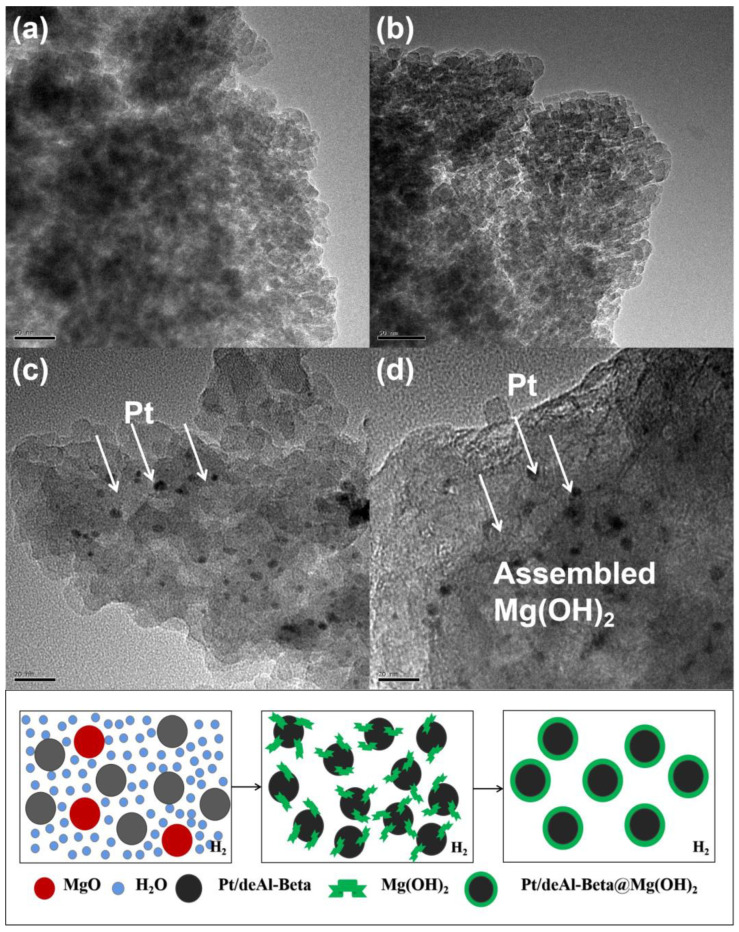
The TEM images of (**a**) Beta; (**b**) deAl-Beta; (**c**) Pt/deAl-Beta; (**d**) Pt/deAl-Beta@Mg(OH)_2_.

**Figure 8 nanomaterials-12-03771-f008:**
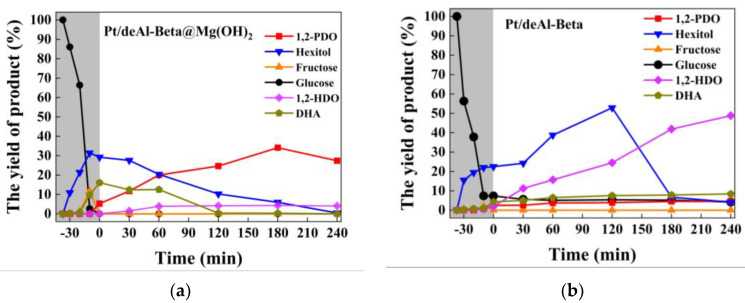
The time course of glucose hydrogenolysis over (**a**) Pt/deAl-Beta@Mg(OH)_2_; and (**b**) Pt/deAl-Beta catalysts.

**Figure 9 nanomaterials-12-03771-f009:**
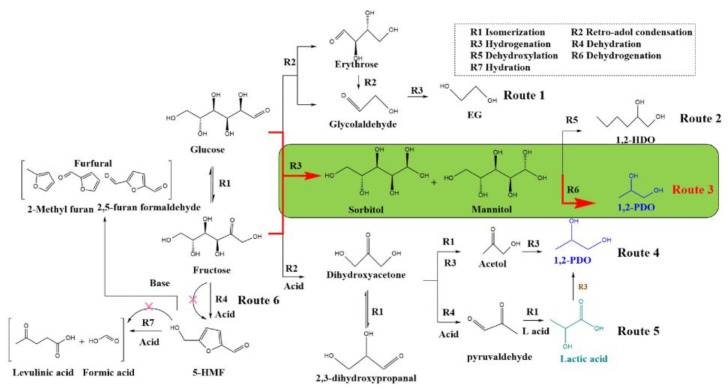
The possible reaction routes of glucose into other products.

**Table 1 nanomaterials-12-03771-t001:** The acid sites of Pt/deAl-Beta and Pt/deAl-Beta@Mg(OH)_2_ catalysts.

Catalyst	Temperature °C	B Acid (m mol/g)	L Acid (m mol/g)	Total Acid (m mol/g)	L Acid Rate L/Total %
Pt/deAl-Beta	150	0.02	0.43	0.45	95.56
	250	0.01	0.28	0.28	100.00
	350	0.00	0.04	0.04	100.00
	450	0.00	0.02	0.02	100.00
Pt/deAl-Beta@Mg(OH)_2_ *	150	0.02	0.38	0.38	95.00
	250	0.01	0.21	0.21	95.45
	350	0.00	0.08	0.08	100.00
	450	0.00	0.02	0.02	100.00

* The Mg(OH)_2_ loading is 7.5 wt%.

## Data Availability

Not applicable.

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
