# Peer review of "Glucose Hydrogenolysis into 1,2-Propanediol Using a Pt/deAl@Mg(OH)2 Catalyst: Expanding the Application of a Core–Shell Structured Catalyst"

_nanomaterials, 2022, doi:10.3390/nano12213771_

Round 1

Reviewer 1 Report

The manuscript presents some interesting work. The following concerns need to be addressed before publication. 

1. Catalyst synthesis details are lacking and need to be improved. Some of the details are ambiguous, which make the procedure could not be duplicated. 

2. Acid-base analysis should also be presented in a figure. 

3. The claims by the authors seem rather speculative. The mechanism is not supported by in situ measurements, or even ex situ experiments such as FTIR. 

4. I would also suggest separating the discussion into their respective sections with the results to allow better visualisation of the data being discussed. 

5. The authors should address the issue of "optimise" yield. Why is the yield at a maximum after 3 hours? What is the reaction pathway after the targeted products? 

Reviewer 2 Report

The paper is remarkably interesting beside concerns a topic of extreme importance for industrial processes, as the authors also declare.

Some changes are suggested to demonstrate the actual improvement of the process compared to those reported in the literature, also because in this case Pt is used which is one of the critical materials catalysts and therefore compromises its industrial development.

Therefore, it would be useful in the introduction to add a table to see such a comparison.

In the experimental part the catalysts have been studied separately it would be useful to use the DOE, that not only allows to verify the optimal conditions for each parameter, but also the possible interaction between the variables, I would suggest to study the variables Pt and Mg(OH)2, just to see any synergistic effects and therefore possibly reduce the amount of Pt or confirm it.

Finally, a revision of English would be useful

Reviewer 3 Report

This paper is concerned with the efficient conversion of glucose to 1,2-propanediol using Pt/deAl@Mg(OH)2 as a catalyst. The efficient conversion of 1.2-PDO using biomass rather than petroleum is a valuable reaction. The authors have already achieved the conversion of glucose to lactic acid using Zn-Sn/β zeolilte catalysts. The present paper describes the conversion of glucose to 1,2-PDO via a number of reactions using Pt/deAl-Beta@Mg(OH)2. The experiments are detailed in terms of the effectiveness of the dealumination,and the loading of Pt and Mg(OH)2. The paper also shows that the catalyst can be reused about three times. The plausible conversion pathway from glucose to 1.2-PDO using this catalyst is also proposed.

Thus, although the contents of the report are deemed worthy of publication, the numbers and legends on the vertical and horizontal axes of all graphs and the chemical structural formulas are too small and unclear, and should be corrected.

If the authors emphasize the effective use of biomass, it would be preferable to mention what the results would be if cellulose were used instead of glucose.

Round 2

Reviewer 1 Report

The authors have addressed the comments adequately, with the exception of the synthesis method. In L107, the authors should attempt to clarify the amount/ratio of catalyst, water, and MgO precursor, and specify what precursor was used for MgO, as the type of precursor used could have unknown effects on the final product. I agree for publication after the authors clarify this. No further review is required. 

Reviewer 2 Report

the changes made are sufficient

Author Response

Dear Nanomaterials editor and reviewers:
Thank you very much for giving us the opportunity to revise the manuscript. Your valuable comments are really helpful to improve our manuscript. We revised the manuscript in accordance with your comments, and checked it carefully to minimize the grammatical errors. We polish our manuscript with your suggestions.